# Modular Piezoresistive Smart Textile for State Estimation of Cloths

**DOI:** 10.3390/s22010222

**Published:** 2021-12-29

**Authors:** Remko Proesmans, Andreas Verleysen, Robbe Vleugels, Paula Veske, Victor-Louis De Gusseme, Francis Wyffels

**Affiliations:** 1IDLab-AIRO, Ghent University, Interuniversity Microelectronics Centre (IMEC), Technologiepark-Zwijnaarde 126, 9052 Zwijnaarde, Belgium; andreas.verleysen@ugent.be (A.V.); robbe.vleugels@ugent.be (R.V.); victorlouis.degusseme@ugent.be (V.-L.D.G.); francis.wyffels@ugent.be (F.W.); 2Centre for Microsystems Technology (CMST), Ghent University, Interuniversity Microelectronics Centre (IMEC), Technologiepark-Zwijnaarde 126, 9052 Zwijnaarde, Belgium; paula.veske@ugent.be

**Keywords:** smart textile, deformable object classification, deformable object manipulation

## Abstract

Smart textiles have found numerous applications ranging from health monitoring to smart homes. Their main allure is their flexibility, which allows for seamless integration of sensing in everyday objects like clothing. The application domain also includes robotics; smart textiles have been used to improve human-robot interaction, to solve the problem of state estimation of soft robots, and for state estimation to enable learning of robotic manipulation of textiles. The latter application provides an alternative to computationally expensive vision-based pipelines and we believe it is the key to accelerate robotic learning of textile manipulation. Current smart textiles, however, maintain wired connections to external units, which impedes robotic manipulation, and lack modularity to facilitate state estimation of large cloths. In this work, we propose an open-source, fully wireless, highly flexible, light, and modular version of a piezoresistive smart textile. Its output stability was experimentally quantified and determined to be sufficient for classification tasks. Its functionality as a state sensor for larger cloths was also verified in a classification task where two of the smart textiles were sewn onto a piece of clothing of which three states are defined. The modular smart textile system was able to recognize these states with average per-class F1-scores ranging from 85.7 to 94.6% with a basic linear classifier.

## 1. Introduction

In [1], smart textiles are defined as textiles that are able to sense stimuli from the environment, to react to them, and to adapt to them by integration of functionality in the textile structure. The stimulus and response can have electrical, thermal, chemical, magnetic, or other origins. This definition encompasses an exceedingly broad range of smart textile systems and corresponding applications. In a medical context, electrodes can be integrated in clothing to monitor electrical activity in the body manifesting as electromyographic (EMG) [2], electrocardiagraphic (ECG) [3], or electroencephalographic (EEG) [4] signals, or to induce electrical currents as a form of electrotherapy [5]. Another example is KneeHapp [6], a system to oversee the rehabilitation of knee injuries that is able to measure muscle shake, pressure on the foot, and bending angles of the leg using inertial measurement units and piezoresistive sensing. Unobtrusive health monitoring can also benefit healthy persons working dangerous jobs: Ref. [7] presents a smart textile-based protective system fully integrated into a firefighter protective suit. The suit is able to monitor heart rate, to detect movements of a firefighter, to detect toxic and combustible gases in the environment, and to measure temperature and relative humidity inside and outside of the suit. Likewise, monitoring of vitals can be highly beneficial in sports. In [8], the heart rate and respiratory frequency of archers is measured using a textile-integrated fiber Bragg grating, enabling archers to better learn how to control these bodily functions and improve their accuracy. Along with the growing interest in these on-body sensor nodes, a new research area has arisen, namely that of wireless body area networks (WBANs) [9]. In a WBAN, both on- and off-body nodes communicate wirelessly to fulfill any of the purposes outlined above. To facilitate this, highly specific conductive textile structures are used as on-body antennas [10] or electromagnetic waveguides [11]. Finally, all applications mentioned up to this point can commonly be described as *wearables*, but use for smart textiles is also found beyond this category; pressure sensitive smart carpets are used for gait monitoring [12] or fall detection [13], employing optical or piezoresistive technologies respectively, and capacitive sensing is integrated in curtains so that they can be controlled at the wave of a hand [14].

In other work [15,16,17,18,19,20,21], smart textiles have additionally been used for classification tasks in the field of robotics. In [18], flexible piezoresistive pressure sensors similar to the one in [6] are applied to a robotic gripper, enabling it to recognize different objects by grasping them. The same sensing principle is explored in detail in [21], with a specific focus on cross-talk cancellation and with intended applications in soft robotics. Soft robots are inherently under-actuated due to their compliance, i.e., they have more degrees of freedom (DoF) than can be controlled [15]. As a result, it is a challenge to identify the state of such a robot for closed-loop control. Multiple researchers have proposed solutions in the form of a smart robot skin [15,16,17]: Ref. [15] uses stretchable parallel-plate capacitors to reconstruct the pose of a 1-DoF pneumatic arm, Ref. [16] integrates both actuators and capacitive sensors in the skin, turning a polymer cylinder into an actuated joint of which both deformation and stiffness can be reconstructed, and in [17] the latter system is enhanced with a learned controller. A different use for a smart textile robotic skin is proposed in [20]; the skin enables robots to recognize different touch gestures and thus improves human–robot interaction.

In the same way that state estimation for soft robots is challenging, so is state estimation of cloths under robotic manipulation. In our previous work [19], we developed a piezoresistive smart textile that is able to classify its own deformation state and showed that an inexpensive dual robotic platform can learn to fold a textile piece in the real world with neural fitted Q-learning and without reward engineering. This approach provides an alternative to computationally expensive vision-based pipelines [22,23]. While [19] presented a valid proof-of-concept of smart textile state estimation for robotic learning of textile folding, the textile used was not suitable for application to larger cloths due to its limited flexibility and the footprint (size and weight) of the readout electronics. In order to reduce the weight and the size of the smart cloth, wired connections are often used to communicate with a signal processing unit separate from the sensor substrate (e.g., [16,17,18,20,21]). When capacitive sensors are used [15,16,17], the capacitance readout is indeed integrated on the textile substrate, but in [16,17], this information is then relayed to a bulky, external Arduino Uno using a wired connection. We argue that a smart textile for state estimation of cloths under robotic manipulation should not have any external wired connections, as these will greatly impede the robot. Furthermore, multiple areas of a cloth can exhibit deformation relevant to its state. Hence, multiple smart textiles may be required. This imposes a need for modularity in the smart textile system. In this work, we propose an open-source, fully wireless, highly flexible, light, and modular version of the piezoresistive smart textile. We demonstrate its viability as a state sensor for larger cloths and aim to use this functionality to accelerate robotic learning of textile manipulation.

## 2. Smart Textile

### 2.1. Structure and Working Principle

In [19], an accessible, low-cost smart textile was developed to solve the problems associated with using vision for the state estimation of cloths. By integrating a flexible tactile sensor grid into a flat textile piece, a pose estimation model can be trained. Consequently, the textile is able to sense and react to its surrounding environment. In this work, the smart textile is refined. Its new composition is shown in Figure 1.

The working principle is the same as in [19]: a sheet of piezoresistive material (Velostat™) is sandwiched between two orthogonal sets of conductive strips (Shieldex^®^ Balingen, < 0.6 Ω/□). This way, a matrix of variable resistors is obtained. Their values change with applied pressure, which is detected by a custom-built printed circuit board (PCB) attached to the textile: one set of strips is connected to digital I/O pins (D${}_{m}$) on the PCB, the other set to analog I/O pins (A${}_{n}$). Obtaining a pressure dependent output from the D${}_{m}$A${}_{n}$ matrix point is done by setting D${}_{m}$ high and reading the voltage $V_{m,n}$ over a fixed pull-down resistor $R_{A}$ at A${}_{n}$. This voltage is related to the pressure dependent resistance $R_{m,n}$ of the D${}_{m}$A${}_{n}$ matrix point through the voltage divider formula:(1)$$V_{m,n}=\frac{R_{A}}{R_{A}+R_{m,n}}\;\cdot \;3.3\;V$$
where 3.3 V is the logical high level of the digital pins. For a full textile readout, all digital pins are sequentially pulled high, and each time a digital pin is high, all analog pins read the voltage over their pull-down resistor. In contrast to [19], the readout is performed by a custom PCB and the textile is constructed as a pouch such that the sheet of Velostat™ can easily be swapped out.

Figure 2 gives an overview of the manufacturing process of the textile pouch. Figure 2a shows the bottom (left) and top (right) layers of the pouch. The conductive strips are laminated onto the textile using the following procedure. Each lamination step uses a Hotronix^®^ Air Fusion IQ^®^ heat press at 170 °C and 4 bar for 40 s. A sheet of Shieldex^®^ Balingen is laminated onto a sheet of 100 μm thick thermoplastic polyurethane (TPU). The combined sheet is cut into strips, which are laid on the textile, TPU side down, and the whole is laminated again. Then, conductive thread (Agsis™, < 0.9 Ω/cm) is sewn at the ends of the strips and towards one of the corners of the textile. When using a sewing machine, the conductive thread should be the bottom wire, while the top wire can be any non-conductive thread. For the top layer, the threads are manually pulled through to the other side of the textile piece after sewing. To ensure that these threads do not unravel, a piece of TPU is laminated over the top. At this point, the top layer is flipped over onto the bottom layer and the loose conductive thread ends of the bottom layer are pulled through the top layer, see Figure 2b. All of this is laminated once more, such that the TPU protecting the conductive threads now also joins the textile layers together. As such, a pouch is obtained with two out of four sides fused together and the other sides left open. In Figure 2c, pushbuttons are sewn near the open ends in between matrix points so that the Velostat™ sheet to be placed in the pouch makes contact with the conductive strips of the textile layers. These buttons can simply be pushed through the Velostat™ without having to make holes in it. The size of the Velostat™ sheet must be such that it entirely covers all of the conductive strips; here it is 17 cm by 17 cm. The custom PCB with a 160 mAh single cell LiPo battery soldered to it is attached to the textile via a 3D-printed holder produced with a Prusa i3 MK3 printer. Through-hole connections are provided on the PCB for attachment of the conductive threads. Figure 2d shows how this is done in detail; the wire is threaded through the hole a few times and standard SMD solder paste is applied. The solder paste is heated using a CIF 852 hot air gun at 250 °C while protecting uncovered parts of the wire with a piece of heat resistant material, e.g., Kapton^®^.

### 2.2. Custom PCB

Figure 3 shows an architecture diagram of the PCB integrated on the smart textile. It is a modular design, meaning multiple smart textiles can be attached to, e.g., the same piece of clothing. This piece of clothing can then feature a single central control unit as in Figure 3a, consisting of a micro-USB port (or any other 5 V source port), a ground, charge and enable wire branching to all PCBs, and a switch to short enable to ground. Plugging a phone charger into the port will charge all PCBs, while shorting the enable wire to ground will turn all PCBs off simultaneously.

The diagram for a single PCB is depicted by Figure 3b. If the central control port is plugged in, 5 V is applied to a charge management controller (MCP73831T-2ACI/OT) to safely charge a single cell 160 mAh LiPo battery powering the PCB. The 5 V line is also used to both power the rest of the circuit and electronically disconnect the battery from the circuit via a rectifier diode (SBR1A40S3-7) during charging. The next stage is a voltage regulator (AP2112K-3.3TRG1) outputting 3.3 V and featuring an enable input. This input is pulled up to the regulator’s supply voltage by default, but if it is shorted to ground, as can be done in the central control unit of Figure 3a, the regulator shorts its output to ground as well. Right after the voltage regulator, a series connection of a Zener diode (BZT52B2V4-E3-18) and a resistor serves as a low battery indicator. If the battery voltage drops below 3.5 V, the voltage regulator will no longer be able to maintain a 3.3 V output. In this case, it outputs its input voltage. With the voltage over the series connection dropping, the Zener diode maintains its Zener voltage of 2.4 V while the voltage over the resistor drops. This drop is detected by the microcontroller unit (MCU), the heart of the PCB: a Nina B306 module. The Nina B306 extends the Bluetooth Low Energy (BLE)-enabled nRF52840 with the required crystal oscillators and a PCB antenna. The nRF52840’s BLE capabilities are used to transmit textile readout values to a remote machine. The module has eight analog I/O pins, of which seven are routed to a through-hole connection to be used for interfacing with the textile. These pins have pull-down resistors connected to them on the PCB; a change in resistance of a Velostat™ matrix point results in a change of voltage over the pull-down resistor. The optimal value for these resistors is such that the voltage swing over them is maximized for different pressure states of the Velostat™ material. The optimal value equals $\sqrt{R_{min}R_{max}}$, with $R_{min}$ and $R_{max}$ the minimal and maximal resistance of the material, respectively. This is proven in Appendix A. To optimize the smart textile as a touch sensor, the resistance of the Velostat™ material is measured at rest ($R_{max}$) and while pressed down with a finger ($R_{min}$), obtaining a value of 1.5 kΩ for the resistors. However, as $R_{min}$ typically has a higher value in a bending application than in a touch application, 3.9 kΩ is a more appropriate value for the former case, and thus for our intended application as deformation state estimator. The eighth analog nRF52840 pin is dedicated to the low battery indicator circuit.

### 2.3. Software

The PCB is Arduino compatible and uses the Arduino Mbed core as developed for the Arduino Nano 33 BLE, which also features a Nina B306 module. Even though the Mbed core uses a 10-bit analog read resolution (the ADC on the Nina B306 supports up to 14-bit resolution), the PCB firmware uses 8-bit values in the interest of throughput. Furthermore, the problem of cross-talk between sensor points as described in [21] is largely solved in software by setting the inactive digital pins to INPUT in Arduino code, which translates to a high-Z state, instead of setting them to zero voltage. This way, inactive digital pins do not sink current from the active pin. Cross-talk can still occur via current-sinking to analog pins other than the active one, which could be solved in hardware using a multiplexer, but this effect is much less prevalent than the previous one. Lastly, programming is done using the serial wire debug protocol. To this end, SWDIO, SWCLK, RESET, GND, and V${}_{\mathrm{DD}}$ (i.e., the source voltage of the Nina module, not that of the entire PCB) pads are provided on the bottom side of the PCB.

### 2.4. General Overview

Table 1 summarizes the specifications of the new smart textile. The integrated PCB is a 20.2 mm wide square and features the Nina B306 MCU. To determine the readout frequency, a remote laptop is set up as a BLE client using the Bleak Python library. When this client is subscribed to notifications on the data characteristic provided by a running smart textile, it receives 66 full textile readouts—with one 8-bit value for each of the 49 matrix points and one for the low battery indicator—per second. Normal operation current consumption is measured by powering the PCB not yet attached to the textile using an Agilent E3631A DC power source and noting the current reading. The power-down current consumption is that of the voltage regulator, as a power-down of the PCB entails a disabling of this component. The battery life was experimentally verified by switching on a finished smart textile and having a remote machine continuously read its data over BLE. The lower bound of the given ranges indicates when the low battery circuit triggers; the upper bound indicates when the PCB shuts down altogether. This latter case occurs when the under-voltage protection circuit on the battery activates. If the battery does not feature such a circuit, further voltage drop after triggering of the on-PCB low battery circuit should be avoided for safety reasons. Should it be desired for the low battery circuit to trigger earlier, this can be achieved by adjusting a simple threshold value in the firmware. In a folded state (see Figure 4 further on), more current flows through the textile, resulting in lower battery life than in a flat resting state.

A cost calculation for one smart textile is given in Table 2. The calculation does not include PCB manufacturing or waste sections of the sheet materials (for example, Velostat™ is often bought in sheets of 28 cm × 28 cm; these fit only one piece of 17 cm × 17 cm as needed for the smart textile).

## 3. Experimental Validation

### 3.1. Pressure Characteristic

In other works [6,18,20,21], piezoresistive matrix-type sensors have been used as normal pressure sensors. Figure 5 shows the output characteristic of a single matrix point of the smart textile in this work under varying pressure. The experiment was performed using a Lloyd LS5 test machine with a 20 N load cell pressing a 2 cm by 2 cm flat square onto one of the textile matrix points with a force of up to 2 N, maintaining every force step for 20 s to let both the testing machine and the textile stabilize. For Figure 5a, 1.5 kΩ resistors are used for pull-down of the analog pins of the readout PCB. Figure 5b, on the other hand, was obtained with 3.9 kΩ pull-down resistors, making the smart textile more sensitive to low pressures, such as those induced by folds rather than external forces. For the selection of the pull-down resistor values, refer to Section 2.2, near the end of the second paragraph.

### 3.2. Stability over Time

Many sensors exhibit some form of drift or degradation in their output values while in use. It follows that the same sensor input may not correspond to the same output when measured at different times. This can be detrimental for applications like classification tasks, where a data set of input–output pairs is collected and each output is assigned a label, with the aim of deriving a model that can predict the label of new inputs. For this reason, an experiment was performed to quantify the stability of the smart textile output values over different iterations of the same task.

One corner of the textile was attached to the end effector of a UR3e collaborative robot. While the textile data was continuously transmitted to a remote laptop over BLE for logging, the robot repeatedly folded the textile diagonally in the exact same way. Every time the textile was fully laid open or fully folded, the robot paused for three seconds. During these intervals, the robot also set one of two digital pins HIGH, one indicating open states, the other folded states. These pins were read by an Arduino MKR1000 that relayed this information to the same remote laptop over a serial connection. The laptop timestamped both the textile and robot pose data. As such, every textile sample can with certainty be attributed to either an open or a folded state. Over the course of 48 h, 10,000 such folds were performed.

Figure 4 shows the data obtained from the output stability experiment. An example of the textile outputs $V_{m,n}$ corresponding to matrix points D${}_{m}$A${}_{n}$ (see Equation (Equation 1)) in a folded state is shown in Figure 4a. Figure 4b shows the textile sensor values obtained for all folded states over the course of the experiment. On average, 5.9 samples were received per iteration during the 3 s interval where the robot paused. For five specific matrix points, raw data is shown in solid black. These were selected based on their mean output values such that the entire range of output values is represented. The remaining, more faint curves are 1000-sample averages corresponding to the data obtained for the other matrix points. For each of the five raw data sets, five empirical cumulative distribution functions (CDFs) were calculated over adjacent iteration intervals; see Figure 4c. The color of each CDF curve indicates the iteration interval in Figure 4b to which it corresponds. Lastly, Figure 6 shows how the textile output varied between open and folded states for the same five prominent matrix points in Figure 4b.

### 3.3. State Classification of Cloths

To demonstrate the use of the smart textile as a state sensor for cloths, a classification task was performed. Two smart textiles were sewn onto a pair of shorts and three states of the shorts were recorded over different iterations: “open” and “folded” states as shown in Figure 7, alongside a “random” state where the shorts were randomly crumpled or dropped on the table. For some samples, the shorts were laid upside down with respect to the depictions in Figure 7a,c. The “folded” state comprises four variants: the first is shown in Figure 7c, but if, for example, the right pant leg is laid over the left, in contrast to Figure 7b, and the lower half of the pant legs is then folded up, the green textile visible in Figure 7a is folded in the opposite way than in Figure 7c. The other two variants arise by placing the shorts face down in the “open” state, as opposed to Figure 7a where they are facing up, then folding either the left half over the right or the right over the left, and again folding the lower half of the pant legs up. Seven different persons collected 45 samples each: 15 open, 15 folded, and 15 random states. During data collection, no visualization of textile data was shown such that the person folding would not be tempted to correct possible uncommon data profiles. To induce further variability in these samples, the shorts were either laid on a flat table or on an uneven surface made up of folded sweaters.

The classifier was trained using logistic regression with l2-norm regularization. The regularization parameters were optimized using cross-validation by performing a search over the logarithmic range [0.0001, 1000] and by using the class-average F1-score as a performance metric. This was done seven times, each time with the data from six out of seven persons in the train set, the remaining as test set. For each test set evaluation, a normalized confusion matrix was computed. Table 3 shows the average and standard deviation of the seven confusion matrices. The average F1-scores on the test sets are 94.6 ± 5.7%, 90.4 ± 4.2%, and 85.7 ± 9.1% for the open, folded, and random classes, respectively.

## 4. Discussion

In this work we present an open-source, fully wireless smart cloth. The smart textile offers several improvements over the first version in [19]. Enhanced flexibility is obtained by employing conductive threads (Agsis™) and sheets (Shieldex^®^ Balingen) instead of the more rigid copper tape and electrical wires used in [19] to implement the conductive matrix. The custom PCB built for the textile is almost four times smaller and comparably lighter than the Arduino MKR1000 used for read out in the first version. Lastly, the textile and conductive layers are connected to form a pouch so that the piece of Velostat™ can easily be swapped out. This is to account for any possible degradation of the Velostat™ material, e.g., creases due to pinching or tight folds. Even though highly specialized materials are used, manufacturing a smart textile is cheap, given that the needed tools and machines (heat press, 3D printer, hot air gun) are available.

First, the smart textile was characterized as a normal pressure sensor. This was done using two different values for the analog pull-down resistors: $R_{A}$, 1.5 kΩ and 3.9 kΩ. As described in Section 2.2, 1.5 kΩ maximizes the output voltage swing over the entire pressure range in which the Velostat™ material exhibits variable resistance. This is desirable for a normal pressure sensor. The obtained characteristic in Figure 5a, however, shows considerable hysteresis; the highest hysteresis value is found for a pressure of 1.5 kPa, with the textile output value with diminishing pressure being 73.9% higher than the output value with rising pressure. Figure 5b, on the other hand, was obtained with 3.9 kΩ pull-down resistors, making the smart textile more sensitive to low pressures such as those induced by deformations rather than external forces. Furthermore, hysteresis is greatly reduced by this compression of the characteristic; maximal hysteresis is found at 0.5 kPa, where the output value with diminishing pressure is 11.3% higher than with rising pressure. For these two reasons, a value of 3.9 kΩ better suits the purpose of state estimation of cloths.

A second experiment aimed to quantify the output stability of the smart textile under repeated identical deformations. In this paragraph, all percentage changes mentioned were measured on 1000-sample averaged curves. Overall, there is no consistent drift in the data for closed states. Some matrix points ended up outputting higher values than at the start (D${}_{6}$A${}_{6}$, D${}_{2}$A${}_{1}$), others output lower values (D${}_{1}$A${}_{0}$, D${}_{0}$A${}_{1}$), or were relatively stable (D${}_{2}$A${}_{2}$). The output value of the D${}_{6}$A${}_{6}$ point increased by 8.94% over the course of the experiment; by iteration 2000, it had already increased by 6.23%. The D${}_{2}$A${}_{1}$ exhibited similar behavior, increasing by 6.55% in the first 2000 iterations and ending with a 9.23% increase with respect to the start. These “setup phases” resulted in much broader CDF curves for the first iteration window; the 10–90 percentile range encompasses 78 mV to 91 mV instead of 26 mV to 39 mV for the other windows. The D${}_{1}$A${}_{0}$ point showed a faster initial change, with a 3.95% decrease in output value after only 300 iterations. For the D${}_{6}$A${}_{6}$ point, a similar change was only reached after 700 iterations. Furthermore, the D${}_{1}$A${}_{0}$ output values were more variable, with even the most stable iteration interval for this matrix point being [2000, 4000], showing a 10–90 percentile range of 52 mV. For the other intervals, the 10–90 percentile range spans between 78 and 91 mV. By the end, the output value had dropped 17.78%. The output of the D${}_{2}$A${}_{2}$ point, on the other hand, only changed by 3.16%. Its 10–90 percentile range encompassed 26 mV to 39 mV. The D${}_{0}$A${}_{1}$ point showed similar stability aside from the [0, 2000] iteration interval, where again a setup phase was observed. At the end of the experiment, its output value had dropped by 27.97%. This is a considerable relative change, though it is inconsequential considering that the mean output value was only 118 mV. In fact, when only considering open states, the highest mean output value for a single matrix point was found to be 472 mV. Hence, any matrix point with a mean output value in closed states lower than 472 mV can be said to have not measured the fold to a significant degree. Out of 13 matrix points exceeding this mean output limit, three showed a start-to-end change of over 10%, one of which being the D${}_{1}$A${}_{0}$ point discussed earlier. Another (D${}_{6}$A${}_{3}$) only just reached the mean output limit, with a value of 488 mV. In short, the smart textile is not to be used for high-precision pressure measurements, but most matrix points show sufficient output stability for classification tasks. This only applies to static states. As Figure 6 shows, the textile output while a fold is being performed is unpredictable; the textile dynamics can cause seemingly random spikes and dips in the readout. However, what happens in the transient regions between defined states is of no concern for the purpose of state estimation.

The applicability for state estimation was demonstrated by bare-bones linear classification of three deformation states of a pair of shorts. Very high classification scores were obtained for the open state, but notable confusion between the folded and random states was observed, along with a high variance of the scores for the random state. The loose definition of what constitutes a random state plays an important role here. Some persons made an effort to thoroughly crumple the short, or were more familiar with the smart textile and hence subconsciously knew how to deform it for accurate classification, while others were more careless and paid no attention to the sensorized regions. This highlights an important parameter in the smart textile classification system, namely sensor coverage. The two smart textiles were strategically placed on the shorts to optimally distinguish open and folded states, but a third may be needed for accurate recognition of the random state.

In [19], it was shown that such state information is invaluable for robotic learning of textile manipulation in a cost-effective and computationally efficient way. With our new enhanced cloth, larger pieces of cloth can be manipulated. The next step is to fuse the tactile information with visual data; in order to detect and grasp the cloth before folding, visual feedback will have to be included. We argue that a multimodal state and reward function estimation exploiting both tactile and visual data will outperform the current state-of-the-art in robotic manipulation of deformable objects [22,23]. 

## Figures and Tables

**Figure 1 sensors-22-00222-f001:**
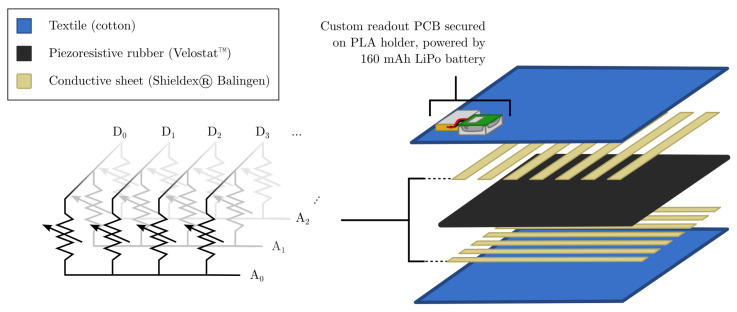
Smart textile structure and principle. The Velostat™ in between two orthogonal sets of conductive strips behaves as a grid of variable resistors.

**Figure 2 sensors-22-00222-f002:**
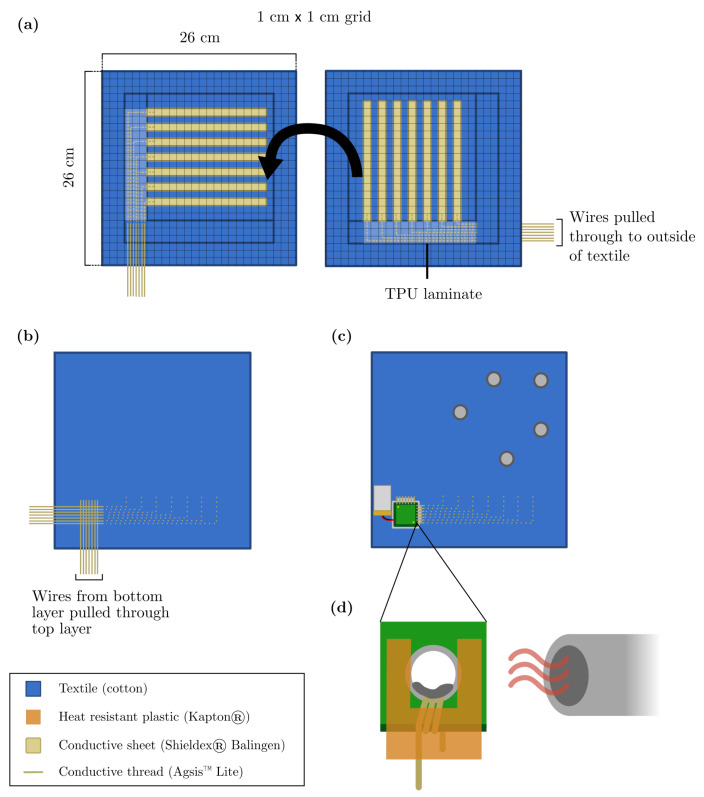
Manufacturing process of the smart textile. (**a**) Dimensions of the bottom (left) and top (right) layer of the textile pouch. The conductive strips are laminated onto the textile using 35 μm thick TPU. Conductive thread is sown onto the strips and routed to a corner of the textile. The black arrow indicates how the top layer should be turned over onto the bottom layer. (**b**) The conductive threads from the bottom layer are pulled through the top layer and both layers are merged by laminating. (**c**) The PCB holder is sewn/glued to the textile and the PCB is attached. The threads are connected to the PCB as detailed in (**d**). Push buttons are sewn into the textile. (**d**) The threads are connected to the PCB by looping them into a through-hole connection, applying standard SMD solder paste, and heating, while protecting the thread itself using a piece of Kapton.

**Figure 3 sensors-22-00222-f003:**
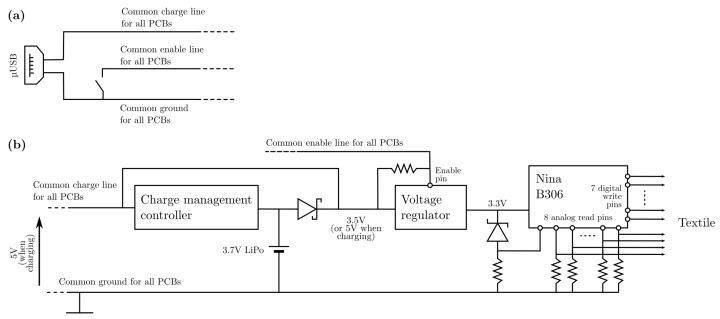
Conceptual schematic of the PCB. (**a**) Central control unit connecting multiple different smart textile PCBs. (**b**) Diagram for one smart textile PCB.

**Figure 4 sensors-22-00222-f004:**
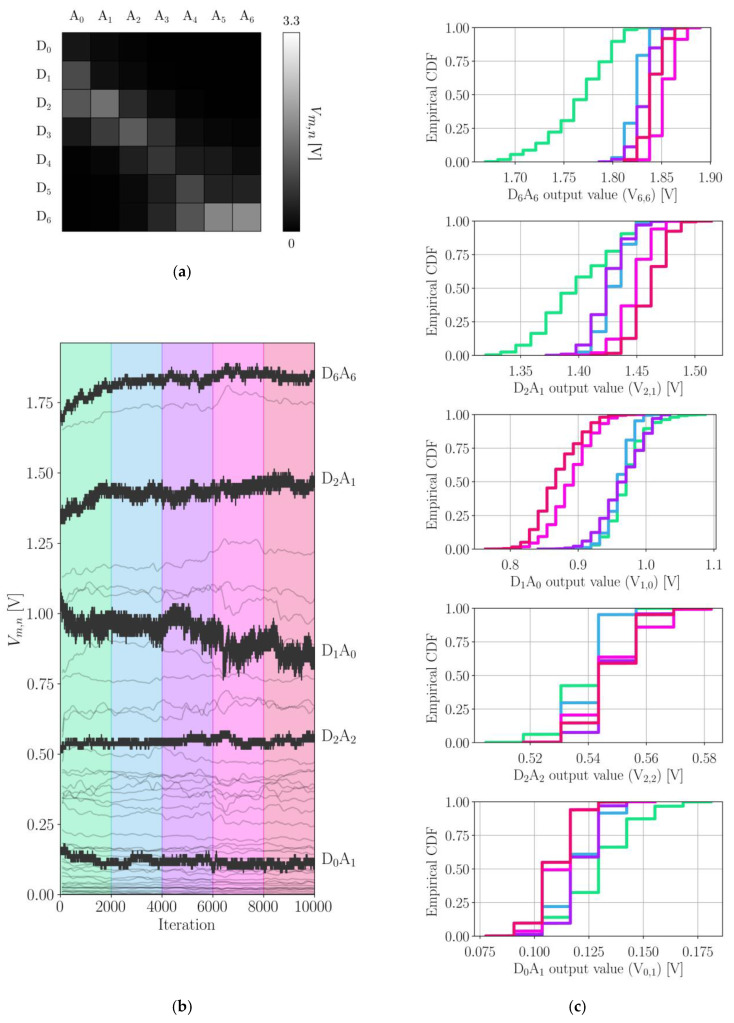
Results of the stability experiment, only including folded states. (**a**) Visualization of the textile output in an example folded state; (**b**) textile sensor values in a folded state over the course of 10,000 iterations. Raw data is shown in black for five matrix points, with corresponding CDFs in (**c**), the other curves are 1000-sample running averages corresponding to the remaining matrix points (**c**). Time-windowed empirical CDFs of the raw data curves are shown in (**b**). Each CDF curve corresponds to an interval in (**b**), as indicated by the coloring.

**Figure 5 sensors-22-00222-f005:**
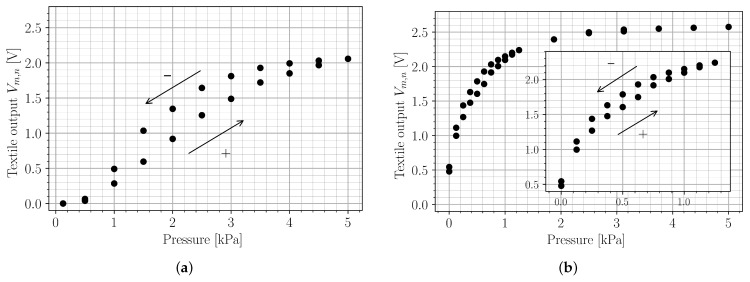
Pressure characteristics of a single smart textile matrix point for different values of the pull-down resistor $R_{A}$ of the analog PCB pins. “+” denotes rising pressure; “−” indicates diminishing pressure. (**a**) $R_{A}=1.5\;$kΩ; (**b**) $R_{A}=3.9\;$kΩ.

**Figure 6 sensors-22-00222-f006:**
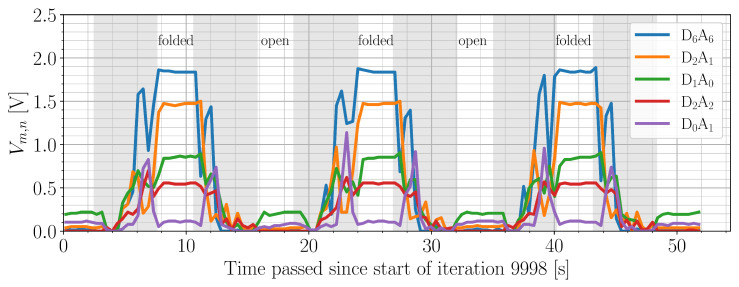
Raw textile output over iterations 9998–10,000 for the five highlighted matrix points in Figure 4b.

**Figure 7 sensors-22-00222-f007:**
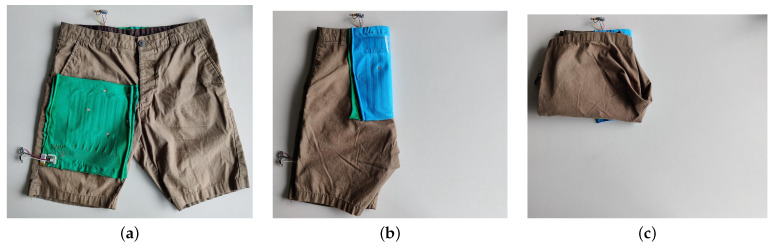
Definition of the “open” and “folded” states of the instrumented shorts. (**a**) “Open” state of the shorts; (**b**) intermediate step between “open” and “folded” states; (**c**) “folded” state of the shorts.

**Table 1 sensors-22-00222-t001:** Technical specifications of the smart textile.

Characteristic	Value
Microcontroller module	Nina B306 with internal nRF52840
PCB size	20.2 mm by 20.2 mm
Readout frequency	66 Hz
Data resolution	8 bit
Normal operation current consumption (PCB only)	7 mA
Power down current consumption	3 µA
Battery life	
At rest	16 h 37 min–16 h 47 min
Folded diagonally	15 h 05 min–15 h 13 min

**Table 2 sensors-22-00222-t002:** Cost calculation for a single smart textile as of October 2021. Prices are based on a certain quantity bought, reported in the respective column. The total price excludes PCB manufacturing.

Part	Qty Needed	Qty Bought	Cost (€)	Source
Nina B306	1	1	11.35	Mouser
Other PCB components	/	1 each	7.57	Mouser
PLA for PCB holder	0.92 g	1 kg	0.04	Mouser
160 mAh LiPo battery	1	5	2.49	EHAO Technology Co., Ltd. (AliExpress)
Cotton textile	1352 cm^2^	1.4 m^2^	0.61	YES Fabrics
Shieldex^®^ Balingen	224 cm^2^	1.4 m^2^	0.70	Shieldex
Velostat™	289 cm^2^	784 cm^2^	1.58	Digikey
Bemis 3914 100 μm TPU	314 cm^2^	68 m^2^	0.31	Bemis Associates Inc.
Agsis™ conductive thread	3 m	30 m	0.67	Syscom Advanced Materials (Amazon)
**Total cost:**	25.32			

**Table 3 sensors-22-00222-t003:** Confusion matrix of the average test scores, normalized rows. All values are percentages.

	open	97.1 ± 4.9	0.0 ± 0.0	2.9 ± 4.9
True Class	folded	1.9 ± 3.0	90.5 ± 7.0	7.6 ± 5.6
	random	6.7 ± 8.7	9.5 ± 7.0	83.8 ± 14.6
		open	folded	random
		Predicted Class		

## Data Availability

Since 22 November 2021, experiment data is available at https://doi.org/10.5281/zenodo.5718426. PCB firmware, board files, and communication code are available at https://github.com/RemkoPr/smart_textile_public.

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
