# Peer review of "Modular Piezoresistive Smart Textile for State Estimation of Cloths"

_sensors, 2021, doi:10.3390/s22010222_

Round 1

Reviewer 1 Report

Ref. No.: sensors-1499709

Title: Modular Piezoresistive Smart Textile for State Estimation of Cloths

In this paper, an open-source, fully wireless, highly flexible, light and modular version of the piezoresistive smart textile was proposed. And the smart textiles were strategically placed on the short to optimally distinguish open and folded states. The article has some new ideas, while some deficiencies in analysis and experiment design should be solved. So, the paper should be further improved by a minor revision.

General comments:

  1. The author should provide the sensing performance of piezoresistive smart fabric to ensure whether it meets the subsequent application.
  2. How can the piezoresistive smart fabric be applied to robotic manipulation?
  3. In page3, line 118, how to determine the use of the 17 cm by 17 cm Velostat™ sheet instead of other specifications?
  4. The working principle of the piezoresistive smart fabric can be described in more detail.
  5. In page 8, line 238, “the short was laid upside down w.r.t. the depictions”, what is the abbreviation of w.r.t. in this paper?
  6. In this paper, the piezoresistive smart fabric response data in the test process can be added.
  7. The author should check the format of references.

Reviewer 2 Report

This paper reports a piezoresistive smart textile, which was improved from their previous work. The paper was well written. The authors should consider the following issues to improve the quality of the paper:

  1. The authors should define “textile output value” in Fig 5(a), i.e, its relationship with the resistance of the sensors.
  2. Because the paper is submitted to Sensors, the authors can consider adding experiment results showing the basic evaluation of the fabricated sensors: the relationship between the resistance changes and the applied force/strain/bending and the effects of temperature change, humidity change.
  3. Raw time series data showing the responses of the sensors when the textile is being folded (changing from open to fold state) should be added to clarify the dynamic response of the sensors.
  4. Please add the comments on the air permeability of the sensors. Also, is the sensor washable?

Round 2

Reviewer 2 Report

The authors have addressed the issues. I recommend accepting the paper for publication.